# Krüppel-like Factor 10 as a Prognostic and Predictive Biomarker of Radiotherapy in Pancreatic Adenocarcinoma

**DOI:** 10.3390/cancers15215212

**Published:** 2023-10-30

**Authors:** Yi-Chih Tsai, Min-Chieh Hsin, Rui-Jun Liu, Ting-Wei Li, Hui-Ju Ch’ang

**Affiliations:** 1National Institute of Cancer Research, National Health Research Institutes, Miaoli 350, Taiwan; yctsai@nhri.edu.tw (Y.-C.T.); sinnhsin@gmail.com (M.-C.H.);; 2Department of Radiation Oncology, Taipei Medical University Hospital, Taipei Medical University, Taipei 110, Taiwan; 3Program for Cancer Biology and Drug Discovery, College of Medical Science and Technology, Taipei Medical University, Taipei 110, Taiwan; 4Department of Oncology, National Cheng Kung University Hospital, College of Medicine, National Cheng Kung University, Tainan 701, Taiwan

**Keywords:** pancreatic adenocarcinoma, radiotherapy, tissue biomarker, Krüppel-like factor 10

## Abstract

**Simple Summary:**

Despite recent improvement in chemotherapy regimens for pancreatic adenocarcinoma (PDAC), the clinical outcomes are still unsatisfactory compared to other solid tumors. Radiotherapy was demonstrated to improve locoregional control of PDAC; however, the survival benefit of radiotherapy in localized PDAC is undefined due to early distant progression in the majority of patients. Upfront chemotherapy for localized PDAC was suggested recently to avoid radical local therapy for patients of localized PDAC high risk of distant metastasis. Potential tissue biomarkers were developed to select PDAC patients who will benefit from local radiotherapy. This review summarizes potential tissue biomarkers reported to predict the efficacy and survival benefits of radiotherapy for localized PDAC including *SMAD4*, a biomarker validated in a prospective clinical trial to correlate with failure pattern of localized PDAC after radiotherapy. In particular, we describe Krüppel-like factor 10 (*KLF10*), lost in two thirds of PDAC patients, in association with distant metastasis and radio-resistance of PDAC. From tumor tissues of patients with resectable PDAC enrolled to a clinical trial, we demonstrated that the combination of *KLF10* and *SMAD4* expression in tumor tissues may help select those who may benefit the most from additional radiotherapy. Though promising, these potential biomarkers should be validated in prospective clinical trials.

**Abstract:**

The prognosis of pancreatic adenocarcinoma (PDAC) remains poor, with a 5-year survival rate of 12%. Although radiotherapy is effective for the locoregional control of PDAC, it does not have survival benefits compared with systemic chemotherapy. Most patients with localized PDAC develop distant metastasis shortly after diagnosis. Upfront chemotherapy has been suggested so that patients with localized PDAC with early distant metastasis do not have to undergo radical local therapy. Several potential tissue markers have been identified for selecting patients who may benefit from local radiotherapy, thereby prolonging their survival. This review summarizes these biomarkers including *SMAD4*, which is significantly associated with PDAC failure patterns and survival. In particular, Krüppel-like factor 10 (*KLF10*) is an early response transcription factor of transforming growth factor (*TGF*)*-β*. Unlike *TGF-β* in advanced cancers, *KLF10* loss in two-thirds of patients with PDAC was associated with rapid distant metastasis and radioresistance; thus, *KLF10* can serve as a predictive and therapeutic marker for PDAC. For patients with resectable PDAC, a combination of *KLF10* and *SMAD4* expression in tumor tissues may help select those who may benefit the most from additional radiotherapy. Future trials should consider upfront systemic therapy or include molecular biomarker-enriched patients without early distant metastasis.

## 1. Introduction

### 1.1. Controversies Regarding Radiotherapy for Pancreatic Adenocarcinoma

Pancreatic adenocarcinoma (PDAC) is notoriously well known for its dismal survival outcomes. It is characterized by rapid distant metastasis or local destructive progression with a 5-year survival rate of 12% [1]. In patients with metastatic or unresectable PDAC, combination chemotherapy regimens consisting of (modified) FOLFIRINOX [2,3] and gemcitabine (GEM) plus nab-paclitaxel [4] have achieved better tumor responses (31.6% vs. 9.4% and 23% vs. 7%, both *p* < 0.001) and overall survival (OS) than single-agent GEM or 5-fluorouracil (11.1 vs. 6.8 months and 8.5 vs. 6.7 months, both *p* < 0.001). Prospective randomized trials revealed the survival benefit of adjuvant chemotherapy with FOLFIRINOX (*p* = 0.003) [5], GEM plus capecitabine (*p* = 0.032) [6], or GEM plus nab-paclitaxel (*p* = 0.009) [7] over single-agent GEM after PDAC resection. For borderline resectable PDAC, neoadjuvant chemotherapy achieved a better R0 resection rate (71% vs. 40%, *p* < 0.01) and survival (15.7 vs. 14.3 months, *p* = 0.025) than upfront surgery [8,9,10].

Unlike chemotherapy, the efficacy of radiotherapy as an adjuvant or curative treatment for PDAC remains controversial. The European Study Group for Pancreatic Cancer (ESPAC)-1 trial indicated no benefit of radiotherapy for resectable PDAC [11]. Our prospective randomized study to revealed that chemoradiotherapy (CRT) with adjuvant GEM for six months improved local control (GEM-CRT arm vs. GEM arm, locoregional recurrence rate: 41.4% vs. 58.1%, *p* = 0.039) but additional CRT had no survival benefit (GEM-CRT arm vs. GEM arm; OS: 21.5 vs. 23.5 months, *p* = 0.82) for patients with resescted PDAC [12]. However, long-term outcomes from the Dutch Pancreatic Cancer Group-initiated PREOPANC study revealed that in patients with (borderline) resectable PDAC receiving adjuvant GEM, neoadjuvant GEM-based CRT had a substantial advantage (5-year OS rate: 20.5% vs. 6.5%, *p* = 0.025) and improved locoregional control (*p* = 0.004) compared with upfront surgery [9,10]. The authors suggested that CRT might benefit patients with PDAC who do not have early distant metastasis. The survival benefit of neoadjuvant therapy for localized PDAC was demonstrated from the ESPAC-5 [13] and the National Clinical Trials Network cooperative groups initiated A021501 [14] trials, especially chemotherapy with FOLFIRINOX in ESPAC-5 showing the 1-year OS rate: 84% vs. 39% (*p* = 0.0028). Neoadjuvant capecitabine-based CRT provided a moderate survival benefit (60% vs. 39%) compared with immediate surgery despite the improved R0 resection and pathologic complete remission rates. The efficacy of neoadjuvant CRT could not be determined in the A021501 trial due to the insufficient accrual of patients after early termination due to the low R0 resection rate in the neoadjuvant CRT arm. The authors concluded that preoperative radiotherapy using other delivery approaches may benefit a subpopulation of patients. Regarding locally advanced PDAC (LAPC), the international LAP07 study identified that the addition of CRT after GEM induction therapy improved local control from 32% to 46% (*p* = 0.03) without survival benefit (*p* = 0.09), partly due to rapid distant metastasis [15]. Conversely, the Eastern Cooperative Oncology Group trial disclosed that upfront GEM-based CRT prolonged median survival duration compared with GEM alone (*p* = 0.017) in LAPC [16]. The conflicting results of randomized studies on localized PDAC imply a narrow therapeutic window for local radiotherapy.

Despite improved clinical outcomes with the combination chemotherapy and neoadjuvant strategy, the survival of patients with PDAC remains inferior to that of patients with other solid tumors [1]. Local recurrence remains one of the essential issues for survival and life quality of PDAC patients. One third of patients with PDAC, disclosed from rapid autopsy, died from local destructive progression without prominent distant metastasis [17]. Despite significant amelioration in recurrence with the mFOLFIRINOX regimen compared with GEM alone in patients with resectable PDAC [5], the pattern of recurrence remained unaffected with isolated locoregional recurrence accounting for 24.6% and 24% of all recurrences in patients who had undergone mFOLFIRINOX and GEM, respectively. A similar observation was noted in the ESPAC-4 study, with local recurrence rates of 53% and 46% in the GEM arm and Gem-CRT arm, respectively [6].

Using personalized radiotherapy, in 49 patients with LAPC, according to the response to induction chemotherapy of eight cycles of FOLFIRINOX and losartan, an inhibitor of thrombospondin-1-mediated activation of latent TGF-β, a phase II trial demonstrated a prominent down-staging and R0 resection rate of 61% with significantly prolonged median progression-free survival (PFS) and OS (17.5 and 31.4 months, respectively) [18]. Ablative radiotherapy following induction chemotherapy with a combination regimen revealed, in 119 patients with inoperable PDAC, safe and prolonged local control, with a median OS of 26.8 months [19]. Advancements in treatment techniques and radiotherapy strategies can be applied to current standard approaches for improving the currently unsatisfactory clinical outcomes of patients with PDAC. Modern radiotherapy provides excellent locoregional control and it should therefore be incorporated into the multimodal treatment of PDAC. Recent clinical trials, especially the PREOPAC study, have implied that administering CRT to patients with a low risk of early distant metastasis can translate local control into survival benefit [4]. Research must be conducted to find patients with PDAC who will benefit the most from CRT by using molecular biomarkers related to PDAC tumorigenesis and progression.

### 1.2. Tissue Biomarkers of Radiotherapy Responses in PDAC

The heterogeneity and aggressive biology of PDAC are classified based on epigenetic, genomic, transcriptomic, and proteomic data [20]. Several potential tissue biomarkers were identified for differentiating progression patterns in patients with PDAC (Table 1). A radiosensitivity index (RSI) for intrinsic radiosensitivity of tumors was developed from a linear regression algorithm of the surviving fraction of 48 cancer cells after 2 Gy and the expression levels of 10 genes including *HDAC1*, *SUMO1*, *PKCb*, *c-Abl*, *STAT1*, *AR*, *Cdk1*, *c-Jun*, *RelA*, and *IRF1*. In 73 patients with PDAC receiving surgery with or without radiotherapy, patients with RSI-high radioresistant tumors tended to have shorter survival (hazard ratio [HR]: 2.1, 95% confidence interval [CI]: 1.0–4.3, *p* = 0.054). For the 31 high-risk patients (positive lymph nodes, positive margins, or postoperative CA19-9 levels > 90 U/mL) who underwent radiotherapy, radio-sensitive patients (i.e., with a low RSI) had significantly improved survival compared with radioresistant patients (i.e., with a high RSI) (OS: 31.2 vs. 13.2 months, *p* = 0.04) [21]. The authors concluded that integrating the RSI with high-risk variables can refine the prognosis of patients with pancreatic cancer treated with radiotherapy. An optimal radiotherapy dose at the individual specific molecular signature level genomic-adjusted radiation dose (GARD) was obtained by combing RSI with a linear-quadratic model. Using data from the total cancer care (TCC) study, the GARD was calculated for 20 primary tumors from various sites treated with the corresponding conventional radiotherapy doses. Despite this uniformity of the radiation dose for a specific tumor type, GARD varied widely across the TCC cohort, implying that a high dose does not always result in a high therapeutic effect. The median GARD was higher in patients with oropharyngeal cancer than in those with non-oropharyngeal head and neck cancer (39.71 vs. 32.56, *p* = 0.042) after 70 Gy; this finding is in concordance with the observation of better efficacy of radiotherapy in patients with oropharyngeal cancer [22,23]. Among the 40 patients in the Moffitt pancreas cancer cohort, the GARD ranged between 16 and 40 and predicted OS independently to a statistically significant level (HR: 2.6, 95% CI: 1.1–6.0; *p* = 0.029). Higher GARDs predicted a better radiotherapeutic effect, longer time to recurrence, and longer survival; moreover, GARD enabled the individualization of the radiation dosage according to tumor radiosensitivity [22,23]. Preclinical studies have demonstrated that indoleamine 2,3 dioxygenase-2 (*IDO2*), a tryptophan catabolic enzyme, promotes pancreatic tumorigenesis. PDAC development reduced in *IDO2^−/−^* mice (30% vs. 10%, *p* < 0.05) [24]. In humans, the high prevalence of two inactivating single-nucleotide variations, rs4503083 [Exon 11] and rs10109853 [Exon 9], of *IDO2* was noted. A DNA analysis of 200 patients from two pancreatic cancer cohorts (The cancer genome atlas and the Thomas Jefferson University Hospital dataset) indicated that an *IDO2*-deficient genotype was correlated with longer PFS in PDAC patients receiving adjuvant radiotherapy (*p* = 0.023). The choline phosphorylation pathway is upregulated in PDAC. From 88 patients with resectable PDAC, metabolic profile analysis demonstrated a prominent difference between good and poor responders in tumors’ choline metabolites (including N-acetylglucosamine-1-phosphate, 1-methylnicotinamide, carnitine, glucose, glutathione, N-acetylglucosamine-6-phosphate, and uridine-5″-monophosphate) regardless of whether they received neoadjuvant CRT. In patients receiving neoadjuvant CRT (n = 62), the levels of carnitine (≤130 nmol/mg), choline (≤283 nmol/mg), phosphocholine (≤749 nmol/mg), and glutathione (≤373 nmol/mg) predicted better PFS (all *p* < 0.05). Multivariate analysis revealed that choline levels of > 284 nmol/mg were significantly associated with recurrence. Microarray analysis confirmed significant suppression of the gene expression levels of the choline transporter *CTL1-4* (*SLC44A1-44A4*) in pancreatic tumor tissues after neoadjuvant CRT. Thus, choline metabolism was suggested as a target and biomarker of neoadjuvant CRT for localized PDAC [25]. Another study integrated genomic profiling and clinical information to predict the radiotherapy response and noted that among 88 patients with cancer receiving radiotherapy, mutations of *CHEK2* (*p* = 0.049), *MSH2* (*p* = 0.014), and *NOTCH1* (*p* = 0.031) were more frequently found in patients with a durable local control of ≥6 months (n = 47). Derangements of DNA repair pathways were associated with better local control (*p* = 0.014). The somatic mutation signature of smoking was observed more often in the durable local control group with a prediction probability of 0.83 for the 6-month local control [26].

*SMAD4* is the only tissue biomarker validated by a prospective trial for predicting failure patterns in PDAC. Among LAPC patients, a local dominant failure pattern was noted in patients with *SMAD4* expression compared to those with *SMAD4* loss (73% vs. 28%, *p* = 0.016) [27]. A retrospective analysis of more than 600 patients with resected PDAC also demonstrated improved survival with adjuvant radiotherapy in only *SMAD4*-positive patients (*p* = 0.002). *SMAD4* loss was significantly associated with metastatic recurrence (HR: 4.28, 95% CI: 2.75–6.68) [28]. *SMAD4* status and expression were correlated with radiosensitivity and PDAC failure patterns in clinical and preclinical studies [17,31,32]. Further studies have demonstrated that the *SMAD4* heterozygous mutation ameliorated PDAC metastatic yet increased its proliferation ability. Loss of *SMAD4* heterozygosity regained PDAC metastatic competency in addition to increased proliferation. Further studies revealed that *RUNX3* interacted with *SMAD4* to modulate cancer cell division and dissemination. This observation implies that a combination of *RUNX3* and *SMAD4* levels can help clinical decision making for resectable PDAC [29].

These tissue biomarkers may optimize the integration of radiotherapy in multimodality treatment for patients with PDAC. Advances in tissue biomarkers facilitate the stratification of patients with PDAC with various potential for distant metastasis and the prediction of those who would benefit the most from additional radiotherapy. With the increasing use of neoadjuvant CRT, especially in borderline resectable PDAC, the value of potential biomarkers in specimens of biopsy, cytology, or peripheral blood should be developed in the future.

Several studies including ours have demonstrated that *Krüppel-like factor* (*KLF*) *10*, a *TGF-β* early-response gene, contributes to radiosensitivity and cancer progression [33,34,35,36]. In the current review, we summarize recent progress in clinical studies of molecular mechanisms of *KLF10* as a predictor of radiotherapy in patients with PDAC.

## 2. Main Text

### 2.1. KLFs

*KLFs* are of the specificity protein 1 (*SP1*)-like/*KLF* transcription factor superfamily and are characterized by the absence of a Buttonhead box, namely CXCPXC [37]. The DNA-binding domain of *KLFs*, located at the carboxyl terminus, contains three conserved C2H2 zinc finger structures. It enables *KLFs* to recognize CACCC elements or GC-boxes and to bind to regulatory regions of the target genes [38]. Eighteen unique members of the *KLF* family were identified, with a >65% sequence similarity for zinc finger motifs, resulting in competition for binding to promoters of target genes (Figure 1). Group 1 consists of *KLF3*, *KLF8*, and *KLF12* which behave as transcriptional repressors by interacting with proteins binding to the carboxyl terminus. Group 2 includes *KLF1*, *KLF2*, *KLF4*, *KLF5*, *KLF6*, and *KLF7* which bind to acetyltransferases and function as transcriptional activators. Group 3 comprises *KLF9*, *KLF10*, *KLF11*, *KLF13*, *KLF14*, and *KLF16* which are transcriptional repressors and interact with switch-independent-3 family member A (*Sin3A*), a common transcriptional corepressor. Nowadays, *KLF15*, *KLF17*, and *KLF18* remain unclassified [39]. *KLFs* are known to be critical regulators of many important biological processes, such as cell proliferation, differentiation, survival, cell cycle, epithelial–mesenchymal transition (EMT), invasion, metastasis, cell maturation, and organogenesis. Dysregulation of *KLF* function can lead to the development of cancer and other disorders [40].

### 2.2. KLF10

*KLF10* was identified in human fetal osteoblasts as a positive regulator of bone growth [41]. The protein homology of *KLF10* among humans, *Mus musculus*, *Bos taurus,* and *Liacerta agilis* is as high as 81.28%, suggesting its critical role in biological processes [42]. *KLF10* is an early-response mediator of *TGFβ/SMAD* signaling. It forms a positive feedback loop with *TGF-β* signaling by transcriptionally regulating *SMAD2* and *SMAD7* [43]. Estrogen stimulates *KLF10* expression, which inhibits *BAX inhibitor-1* transcription and enhances breast cancer cell apoptosis [44]. *Jun B* and *lysine demethylase 6A* may facilitate *KLF10* transcription to exacerbate diabetic nephropathy [45]. Multiple long noncoding RNAs (lncRNA) and microRNAs (miRNA) were identified as upstream regulators of *KLFs*, thus providing essential pathways for targeting *KLFs*. E3 ubiquitin ligases, including seven in absentia homolog-1 (SIAH1) and FBW7, interact with *KLF10* through conserved binding motifs to promote the proteasomal degradation of KLF10. The binding of *KLF10* to itchy E3 ubiquitin ligase (ITCH) increases KLF10 levels and activates Foxp3 transcription in regulatory T cells [46,47]. We previously reported that KLF10 is a phosphorylated protein at Thr-93 in the N-terminal region. RAF-1 phosphorylation and PIN1 isomerization coordinately regulate KLF10 stability and tumor progression [48].

### 2.3. Involvement of KFL10 in Multiple Diseases

*KLF10* is involved in glucose and lipid metabolism, mitochondrial structure and function, cell proliferation, and apoptosis and it plays critical roles in multiple diseases [49]. It is a clock-controlled gene that maintains the hepatic circadian rhythm which is essential for regulating hepatic glucose and lipid homeostasis [42]. Sex-dependent differences were found in the metabolic phenotypes of *KLF10*-knockout mice. Male mice exhibited post-prandial and fasting hyperglycaemia whereas female mice exhibited increased plasma triglyceride levels. As a circadian-clock-controlled transcription factor, *KLF10* suppresses lipogenic genes of glucose and lipid metabolism in the liver and it affects gluconeogenesis, contributing to diabetes [50,51]. *KLF10* alleviates hepatic steatosis and nonalcoholic-steatohepatitis by downregulating *SREBP-1c* involving lipogenesis [52,53]. *KLF10*-deficient mice exhibit reduced receptor activator of nuclear factor *kappa*-B ligand, increased osteoprotegerin, and delayed8 osteoclast differentiation which led to reduced bone turnover and osteopenia [49,51,54]. A study reported that male *KLF-*knockout mice developed cardiac hypertrophy after approximately 16 months due to the angiotensin II-induced cardiac transcription factor, *GATA4*, and the atrial natriuretic factor, brain natriuretic peptide [55]. *KLF10* can transactivate *Foxp3* promoters in regulatory T cells in response to *TG-β1* to promote atherosclerosis [56,57].

### 2.4. KLF10 in Cancer

Many studies have demonstrated the tumor suppressor function of *KLF10* in terms of cell proliferation inhibition and apoptosis induction [58,59]. *KLF10* loss activates PTEN/PI3K/*AKT* activity in multiple myeloma and bladder cancer [60,61]. *KLF10* overexpression can suppress *Wnt* signaling and *GSK3β* phosphorylation to inhibit the proliferation, migration, and drug resistance of multiple myeloma cells. Knock-down of securin, the downstream target of *KLF10*, can mimic the tumor suppressor role of *KLF10* in multiple myeloma [62]. In advanced-stage cancer, *TGF-β* signaling enhances the EMT whereas *KLF10* inhibits *TGF-β*-induced EMT. *KLF10* can suppress lung and pancreatic cancer EMT and invasion by recruiting *HDAC1* to suppress the SNAI2 promoter for the removal of histone acetylation (H3K9ac and H3K27ac) [63]. In oral squamous cell carcinoma, *KLF10* was identified as a differentially expressed circadian-related gene that was correlated with OS (*p* < 0.05) and the drug response (*p* = 0.0014) [64]. By directly binding to the LINC00629 promoter to induce Mcl1 degradation, *KLF10* exerts antitumor activity in oral squamous cell carcinoma treated with apigenin, a flavonoid [65]. *KLF10* is involved in cervical cancer immunoediting by transcriptionally regulating IL6, IL25, and pregnancy*-specific beta-1 glycoproteins 2* and *5* [66]. Conversely, the tumor suppressive role of *KLF10* may vary depending on the tumor cells types and the microenvironments. In *KLF10*-knockout mice, the *TGF-β-SMAD* signaling pathway was activated to suppress diethylnitrosamine-induced hepatocyte proliferation in the liver cancer [67].

### 2.5. Role of KLF10 in PDAC Progression

Studies have revealed associations between PDAC and alterations in *TGF-β* receptor genes and *SMAD* [68,69]. However, no alterations in *KLF10* expression were found in a mutation screening study of 22 human pancreatic cancer cell lines [70]. *KLF10* expression in various cancer tissues has been reported to be significantly lower than that in normal tissues [63,71]. In PDAC, *KLF10* expression was low in two thirds of patients and was inversely correlated with the cancer stage [36,53]. Despite alterations in the *TGF-β* signaling pathway components in patients with PDAC, *KLF 10* could regulate *TGF-β* signaling and inhibit epithelial cell proliferation in pancreatic cancer cells [72]. KLF10 expression can be increased by a noncoding RNA, lncRNA *FLVCR1-AS1*, by acting as a competitive endogenous RNA to sequester the inhibitory effects of miR-513c-5p or miR-514b-5p. Since lncRNA *FLVCR1-AS1* is a direct transcriptional target of *KLF10*, this *FLVCR1-AS1/KLF10* positive feedback loop can suppress PDAC progression [73].

In the murine model of pancreas-specific *KLF10* deletion (*Pdx-1Cre KLF10^L/L^*), no evidence of abnormal pancreas development or neoplastic lesions was noted. The synergistic effects of *KLF10* inactivation-activated mutant *Kras^G12D^* in cross-breed mice led to the rapid onset of advanced PDAC with 50% penetrance. The upregulation of *c-Jun* and *SDF-1/CXCR4* signaling after *KLF10* deletion was responsible for accelerated PDAC cell growth and distant metastasis [74]. Since *KLF10*-knockout mice exhibited a high incidence of metabolic disorders, we previously explored *sirtuin6*, an NAD+-dependent deacetylase downstream of *KLF10*, as a key regulator of glucose homeostasis and a tumor suppressor. Our findings indicated that *KLF10* transcriptionally activated *sirtuin6* to modulate the EMT and glycolysis of PDAC coordinately through *NFκB* and *HIF1α* [75]. In addition to the *Wnt/β-catenin* signaling pathway, we demonstrated that *KLF10* contributed to the cancer stemness phenotype by transcriptionally regulating *Notch-3* and *Notch-4* and competing with *E74-like ETS transcription factor 3* (ELF3) for promoter binding. A combination of metformin, which upregulates *KLF10* by phosphorylating AMP-activated protein kinase, and evodiamine, a nontoxic *Notch-3* methylation stimulator, ameliorated PDAC growth through *KLF10* downregulation [76] (Figure 2).

### 2.6. Role of KLF10 in PDAC Resistance to Radiotherapy

The *KLF* family regulate radiosensitivity in various cancers (Table 2). *KLF2* and *KLF4* are positive regulators of endothelial-protective molecules such as nitric oxide and thrombomodulin. Compared with single-dose radiation, fractionated radiation markedly reduced the *ERK5/KLF2* pathway and enhanced *ICAM-1* expression, leading to endothelial dysfunction [77]. *KLF4* and *KLF5* may prevent radiation-induced intestinal injury by inhibiting apoptosis and modulating DNA repair pathways [78,79]. *KLF4* expression can predict radiotherapy resistance and poor clinical outcomes for cervical cancer. From tumor tissues of 117 patients with locally advanced cervical cancer, *KLF4* was disclosed as a risk factor for radioresistance (*p* = 0.032), poor PFS (*p* = 0.001), and OS (*p* < 0.001) [80]. *KLF5* was the predictor of poor response to CRT in rectal cancer [81]. In colon cancer cells, radiation time-dependently and dose-dependently stabilized *KLF5* levels. *KLF5* increased *cyclin D1* and *β-catenin* levels to mediate cell survival. A study assessing 60 colorectal tumor tissues before radiotherapy indicated that high *KLF5* expression was correlated with pathologic complete remission (*p* = 0.023) and radioresistance in colorectal cancer [81]. High *KLF6* expression level was associated with a nearly four times higher risk of local recurrence in head and neck cancer patients after radiotherapy (*p* = 0.008) [82].

*KLF10* gene expression can be used to discriminate between γ-radiation and α-radiation quality [83]. Radiation-induced delayed neuropsychiatric disorders was associated with biological processes, such as protein kinase activity, circadian behavior, and cell differentiation. The alteration of expression levels of six genes, including KLF10, in the chronic phase of radiation increased anxiety-like behaviors in mice [84]. Radiation-induced *KLF10* upregulation was noted in many cancer cell lines and murine models [83,84,85]. *KLF10* transcriptionally downregulated *EGFR* and modulated gemcitabine-resistance in cholangiocarcinoma [86]. In esophageal squamous cell carcinoma, exosomes secreted from hypoxic tumors after radiation expressed high levels of miR-340-5p, which suppressed KLF10 transcription. Higher miR-340-5p expression and lower *KLF10* expression in plasma exosomes from patients with esophageal cancer patients were associated with poorer radiation responses and prognosis [33]. Several studies, including ours, have demonstrated that *KLF10* transcriptionally suppresses the UV radiation resistance-associated gene (*UVRAG*) and modulates apoptosis, DNA repair, and autophagy in cancer cells. Metformin might decrease radioresistance in pancreatic and esophageal cancers by elevating *KLF10* expression [33,35]. Furthermore, EMT and cancer stem cell phenotypes also contribute to radioresistance [87,88]. *KLF10* modulates EMT and can lead to cancer stemness phenotypes by transcriptionally regulating *sirtuin6*, *Notch-3,* and *Notch-4*, respectively, and thus may cause radioresistance in PDAC [74,75,76]. Whether *KLF* family members share promoter binding sites on *UVRAG* or other signal targets and regulate the balance between radiosensitivity and radioresistance warrants further exploration.

### 2.7. Selection of Patients with Resectable PDAC for Radiotherapy Using KLF10 and SMAD4

To evaluate the benefits of additional CRT to standard adjuvant chemotherapy in patients with resected PDAC, we conducted a randomized clinical trial from 2009 to 2015 [12]. We enrolled 147 patients with PDAC after curative resection and randomized them to either adjuvant GEM 1000 mg/m^2^ infusion weekly for six cycles or adjuvant GEM for three cycles and GEM (400 mg/m^2^ weekly)-based CRT and another three cycles of GEM. Despite the significant locoregional benefit (*p* = 0.039) of additional CRT, the median recurrence-free survival and OS were of no significant difference in the two arms (HR: 0.98, *p* = 0.89 and HR: 1.04, *p* = 0.82), respectively (Figure 3A) [12]. Tumor specimens were collected from 111 patients. Immunohistochemical expression of biomarkers including *KLF10*, *SMAD4*, and *RUNX3* was evaluated by pathologists using a visual grading system based on staining intensity and extent. The postoperative CA19-9 level and protein expression of *KLF10* and *SAMD4*, were significantly associated with OS (*p* = 0.047, 0.013, and 0.045, respectively). High *KLF10* or *SMAD4* expression in patients (n = 55) receiving additional adjuvant CRT had a significantly prolonged local control time (ꚙ vs. 19.8 months, *p* = 0.026) and a better OS (33.0 vs. 23.0 months, *p* = 0.12) than those receiving GEM alone. In resected PDAC patients who had a loss of both *SMAD4* and *KLF10*, additional adjuvant CRT caused the rapid development of distant metastasis and worse clinical outcomes (Figure 3B) [30]. The combination of *KLF10* and CA19-9 levels did not reveal significant differences in survival outcomes between the treatment arms [30]. On the basis of these findings, we concluded that the chances of translating locoregional control of CRT into prolonged survival were high in patients with *KLF10-* or *SMAD4*-expressing tumors. Although these findings are promising, a prospective study is warranted to validate the results.

## 3. Conclusions

The value of radiotherapy in PDAC remains unclear due to conflicting results of clinical trials [89,90,91]. Modern radiotherapy is efficacious, resulting in a satisfactory safety profile and local control for patients with localized PDAC [18,19]. Locoregional control of the primary tumor is crucial for patients with PDAC and is increasingly possible with advancements in chemotherapy [92]. Retrospective studies have identified potential tissue biomarkers for predicting the benefits of enhanced locoregional therapy. However, most candidate biomarkers were only correlated with survival but not with failure patterns. Thus, prospective clinical trials in patients with PDAC receiving modern chemotherapy with or without up-to-date radiotherapy are required to validate the efficacy of biomarkers in selecting optimal patients for radiotherapy.

In preclinical studies, *KLF10* was demonstrated to be correlated with PDAC progression and resistance and it was reported to modulate distant metastasis, cancer stemness, and radio-sensitivity. A retrospective analysis of prospective randomized trials concluded that the combination of *KLF10* and *SMAD4* expression can help select patients with resected PDAC who may be suitable for local radiotherapy. Current enthusiasm of upfront systemic chemotherapy for localized PDAC aims to prevent patients with rapid distant metastasis from radical local therapy including radiotherapy. Future trials evaluating the efficacy of radiotherapy in PDAC should focus on molecular biomarker-enriched patients who carry a low risk of early distant metastasis.

## Figures and Tables

**Figure 1 cancers-15-05212-f001:**
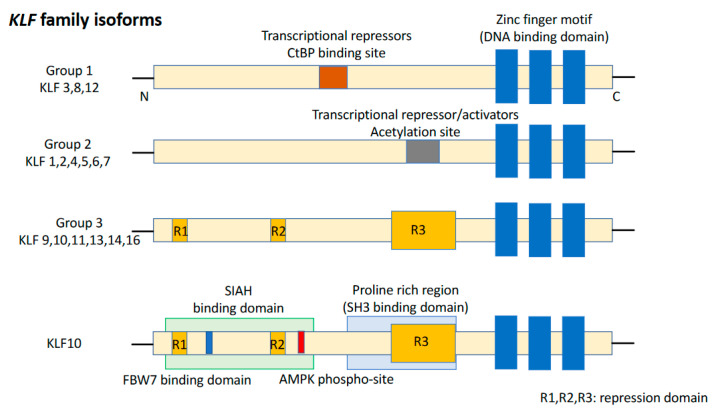
Structure of the KLF family and KLF10. KLF isoforms can be divided into three groups. KLF10 belongs to Group 3. KLF15, 17, and 18 are not included in any of these groups.

**Figure 2 cancers-15-05212-f002:**
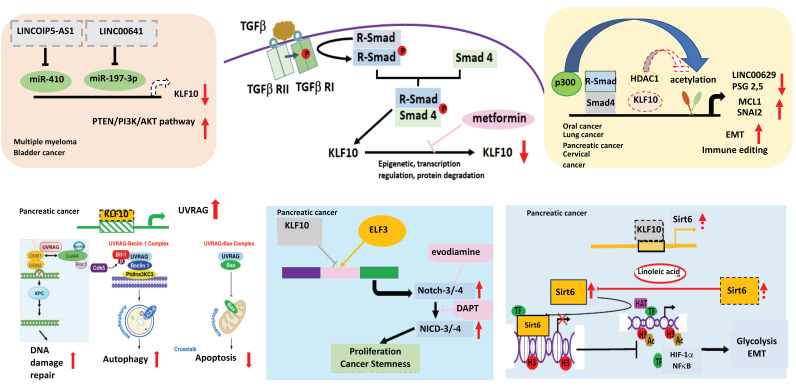
The role of KLF10 in cancers. KLF10 can transcriptionally regulate lnc00629 to modulate MCL1 expression in oral cancer. KLF10 inhibits multiple myeloma (MM) and bladder cancer by increasing the expression of PTEN to reduce AKT activity. In MM, miR410, which can be suppressed by lncRNAOIP5-AS1 inhibits the expression of KLF10. In bladder cancer, lncRNA00641 can suppress miR-197-3p to modulate KLF10 transcription. In cervical cancer, KLF10 regulates PSG2,5 to modulate the tumor immune environment. KLF10 can inhibit the TGFβ-induced epithelial–mesenchymal transition (EMT) to suppress cancer invasion in the lung and pancreas by recruiting HDAC1 to block the expression of SNAI2. In pancreatic adenocarcinoma, KLF10 transcriptionally suppresses the UV radiation-associated gene (UVRAG) to modulate DNA damage repair, autophagy, and the apoptosis of cancer cells. KLF10 competes with *E74-like ETS transcription factor 3* (ELF3) in binding to Notch-3 and -4 promoters to suppress cancer proliferation and the stemness phenotype. KLF10 can transcriptionally activate sirtuin6 to coordinate glycolysis and EMT of pancreatic cancer via HIF1α and NFkB. Upward arrows denotes increase; downward arrows represent decrease. Lines with blunt end means inhibit. Thicker lines represent prominent increase (or decrease) and vice versa.

**Figure 3 cancers-15-05212-f003:**
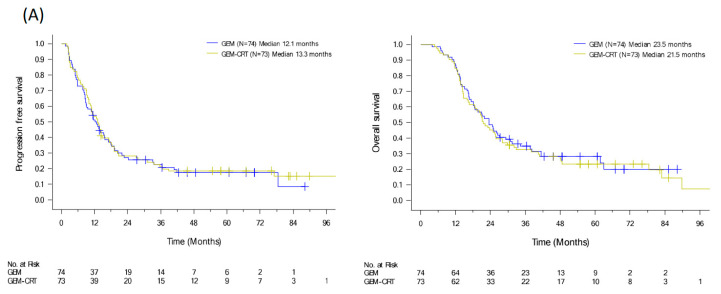
PFS and OS curves of (**A**) 147 resectable pancreatic cancer patients randomized to adjuvant gemcitabine (GEM, n = 74) with or without additional adjuvant gemcitabine-based chemoradiotherapy (GEM-CRT, n = 73). (**B**) In total, 101 patients enrolled to the above mentioned clinical trial with qualified tumor tissues. The levels of KLF10 and SMAD4 were evaluated by two pathologists blinded to the clinical information. Patients with high expression of KLF10 or SMAD4 (H/H, n = 55) should receive additional adjuvant CRT than GEM only due to a significantly better PFS (NA vs. 19.8 months; *p* = 0.026) and a longer OS (33.0 vs. 23.0 months; *p* = 0.12). Conversely, adjuvant CRT after curative resection may not be suitable in those with low expression of both KLF10 and SMAD4 (LL) who develop distant metastasis rapidly.

**Table 1 cancers-15-05212-t001:** Tissue biomarkers to predict radiotherapy responses in pancreatic adenocarcinoma. Representative clinical studies evaluating potential tissue biomarkers in correlation with survival or failure pattern of radiotherapy to PDAC.

Study	Study Type	Pt No.	PDAC Stage	Tissue Origin	Treatment	Biomarker	End-Point	Conclusion	Significance
2015 Strom T [21]	Retro-spective	73	resectable	DNA	Adjuvant GEM/5FU ± RT (n = 61) vs. No adjuvant therapy (n = 12)	10 specific genes (RSI score)	OS	Among clinical high risk irradiated patients, RSI low (radiosensitive) had significantly improved survival	RSI low vs. RSI high OS: 31.2 vs. 13.2 months, *p* = 0.04
2017 Scott JG [22]	Retrospective	40/8271	NA	DNA/Moffit Cohort	Radiotherapy 45–54 Gy	GARD	OS	High GARD is associated with radiosensitive and better clinical outcomes	HR: 2.6; *p* = 0.029
2019 Nevler A [24]	Retro-spective	129	resected	DNA/TCGA	With or without radiotherapy	Indoleamine 2,3 dioxygenase 2 (IDO2)	RFS	*IDO2* inactivation associated with improved RFS in response to RT	*p* = 0.023
2022 Wada Y [25]	Retro-spective	88	resected	Frozen tissue	Resected with or without Neoadjuvant CRT	Choline metabolites	RFS	Reduced choline metabolites correlate with better RFS especially in NA-CRT group	Choline: *p* = 0.0022 (in NA-CRT: *p* = 0.028) Phospho-choline: *p* = 0.0086 (in NA-CRT *p* = 0.0037)
2022 Jang BS [26]	Retro-spective	2/88	NA	DNA	radiotherapy	*CHEK2*, *MSH2*, *NOTCH1*	LFFS	Mutations of *NOTCH2* and *BCL* were enriched in the NDLC group; Mutations of *CHEK2*, *MSH2* and *NOTCH1* were more frequently in the DLC group.	Altered DNA repair pathway was associated with better LFFS (HR: 0.4; *p* = 0.014)
2011 Crane CH [27]	Pro-spective phase II	69	LA	Cytology	GEMOX + cetuximab+ capecitabine-CRT	*Smad4*	Failure pattern	Pattern of progression may be predictable on the basis of *Smad4* expression	intact *Smad4* in 11/15 (73.3%) of local dominant recurrence; *Smad4* loss in 10/14 (71.4%) of distant dominant recurrence. *p* = 0.016
2017 Shin SH [28]	Retro-spective	641	resectable	IHC	Adjuvant 5-FU/LV or GEM; 5-FU-CRT for R1 resection	Smad4	OS, recurrence	1. Inactivation *Smad4* indicate metastasis 2. In expressed *Smad4*, local therapy contributes to improved survival	1. HR: 4.28 2. *p* = 0.002
Iacobuzio Donahue 2009 JCO [17]	Retrospective	76	Stage I/II: 22; III:18, IV:36.	IHC	Surgery, chemotherapy, CRT	*Smad4* *TP53* *Kras2*	failure pattern: local vs. distant	*Smad4* loss in 2/9 (22%) LA without metastasis; 16/22 (78%) with 100–100 of metastases (*p* = 0.032)	*Smad4* expression correlated with pattern of failure (locally destructive vs. metastatic) *p* = 0.007
2015 Whittle MC [29]	Retro-spective	88	resectable	IHC/ICGC	Chemotherapy with or without radiotherapy	*Smad4*, *Runx3*	OS, relapse pattern	Low *Runx3* benefit from radiotherapy. High *Runx 3* and loss of *Smad4* pose the greatest challenge	High *Runx3* correlated with poor median survival (*p* < 0.018).
2021 Pen SL [30]	Pro-spective phase III	111	resectable	IHC	Adjuvant GEM ± GEM-CRT	*Smad4*, *KLF10*, *Runx3*	OS, RFS	Combining *KLF10* and *Smad4* may predict the benefits of adjuvant CRT in resected PDAC	High *KLF10* or *Smad4* (n = 55) had better local RFS (*p* = 0.026) and longer OS (*p* = 0.12) receiving adjuvant CRT than GEM alone.

Pt no.: patient number; PDAC: pancreatic adenocarcinoma; GEM: gemcitabine; 5FU: 5-fluoruracil; RT: radiotherapy; RSI: radiation sensitivity index; OS: overall survival; GARD: genomic-adjusted radiation dose; HR: hazard ratio; RFS: recurrence-free survival; NA-CRT: neoadjuvant chemoradiotherapy; LFFS: local failure free survival; NDLC: non-durable local control; DLC: durable local control; LA: locally advanced; GEMOX: gemcitabine + oxaliplatin; IHC: immunohistochemistry; LV: leucovorin; NA: not available; ICGC: International Cancer Genome Consortium; TCGA: the cancer genome atlas.

**Table 2 cancers-15-05212-t002:** KLFs in regulating the radiation sensitivity of cancers. Representative studies evaluating KLF family members in correlation with clinical outcomes of radiotherapy in various cancers.

Study	Cancer Type	Patient No.	Treatment	Tissue Collection	Analysis	Findings	Mechanisms
2021 Chen F [33]	Esophageal cancer	88	60 Gy (2 Gy/fx) + cisplatin and fluorouracil	Blood, tumor tissue	Histologic and plasma exosomal miR-340-5p and KLF10	Histologic and exosomal miR-340-5p levels correlated with tumor recurrence (*p* < 0.0001, *p* = 0.0004) and overall survival (*p* = 0.0026, *p* = 0.0076); miR-340-5p expression negatively correlated with KLF10	Exosomal miR-340-5p is critical for hypoxic exosomal transferred radioresistance. KLF10 was a direct target of miR-340-5p. Metformin may increase the expression of KLF10 and enhance the radiosensitivity of esophageal cancer
2017 Chang VH [35]	Pancreatic cancer	20	Neoadjuvant 50.4 Gy/28fx + Gemcitabine	Tumor tissue IHC	Histologic KLF10, UVRAG	High KLF10 expression correlated with better tumor regression grade (R = −0.69, *p* = 0.001) KLF10 expression was inversely correlated with UVRAG (R = −0.259, *p* = 0.03)	KLF10 transcriptionally suppressed UVRAG to enhance radiosensitivity via modulating apoptosis, DNA repair, and autophagy
2017 Liu HX [80]	Cervical cancer	117	Radical radiotherapy + brachytherapy + cisplatin-based chemotherapy	Tumor tissue IHC	Histologic KLF4	High KLF4 expression correlated with shorter PFS (*p* = 0.0019) and OS (*p* < 0.0017)	High expression of KLF4 promoted radioresistance. KLF4 induces p21 leading to cell cycle arrest and suppressing BAX expression, thus reducing apoptosis
2019 Kim JY [81]	Rectal cancer	60	Preoperative 50.4 Gy/28fx + 5-FU/LV	Tumor tissue IHC	EGFR, p53, KLF5, C-ern, Ki67	KLF5 expression was a significant worse factor for pCR (*p* = 0.012). Radiation stabilizes KLF5 protein in a time and dose dependent manner	KLF5 increased cyclin D1 and β-catenin to promote cancer cell survival. KLF5 expression depends on Kras and Braf mutations
2021 Leon X [82]	Head and Neck cancer	83	70–72 Gy to primary tumor and 50 Gy on nodal areas in N0 or 70–72 Gy in N1 disease	Tumor tissue RT-PCR	KLF6	High KLF6 expression had a 3.8 times higher risk of local recurrence after radiotherapy (*p* = 0.008)	KLF6 regulates response to cancer therapy in a p53-dependent manner and it promotes tumor progression from the transcriptional activation of TGFβ

Fx: fraction; IHC: immunohistochemistry; UVRAG: UV radiation resistance associated gene; PFS: progression-free survival; OS: overall survival; 5-FU/LV: 5-fluorouracil/leucovorin; pCR: pathologic complete remission.

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
