# Peer review of "Krüppel-like Factor 10 as a Prognostic and Predictive Biomarker of Radiotherapy in Pancreatic Adenocarcinoma"

_cancers, 2023, doi:10.3390/cancers15215212_

Round 1

Reviewer 1 Report

Comments and Suggestions for Authors

I appreciate the authors' work on this review and the points that they reach. I ask they will consider these comments.

First, I thorough proofreading is necessary. There are multiple instances of grammatical errors, words that seem to be left in after vision, and so on.

Second, the authors provide an excellent summary of the controversial role of radiation therapy as adjuvant. The idea to use a biomarker, in this case KLF10, to determine who would benefit from adjuvant RT is interesting. I agree this would need to be explored in study. But, what about neoadjuvant? There is increasing use of neoadjuvant RT, especially for tumors deemed borderline. I recommend including this in their review.

Comments on the Quality of English Language

see my prior comments

Author Response

Reviewer 1

I appreciate the authors' work on this review and the points that they reach. I ask they will consider these comments.

  1. First, I thorough proofreading is necessary. There are multiple instances of grammatical errors, words that seem to be left in after vision, and so on.

Ans: Thank you for the suggestion. The manuscript has been revised by professional English editing service (Wallace editing academic). We have asked a colleague fluent in English writing to -checked the manuscript further.

  1. Second, the authors provide an excellent summary of the controversial role of radiation therapy as adjuvant. The idea to use a biomarker, in this case KLF10, to determine who would benefit from adjuvant RT is interesting. I agree this would need to be explored in study. But, what about neoadjuvant? There is increasing use of neoadjuvant RT, especially for tumors deemed borderline. I recommend including this in their review.

Ans: Thank you for the comments. We agree with you the increasing use of neoadjuvant RT especially in borderline resectable pancreatic cancer. We added a sentence on lane 177 to 180 to address this issue: With increasing use of neoadjuvant CRT especially in borderline resectable PDAC, the value of potential biomarkers in specimens of biopsy, cytology or peripheral blood should be developed in the future.

Reviewer 2 Report

Comments and Suggestions for Authors

The authors of the manuscript entitled "Krüppel-like Factor 10 as a Prognostic and Predictive Biomarker of Radiotherapy in Pancreatic Adenocarcinoma", Yi-Chih Tsai et al have done a commendable job in summarizing the role of SMAD4 and KLF10 in PDAC as well as their association with TGFB. The review however, directs towards the need for future trials to establish such biomarkers. The manuscript is well written and comprehensive. However, there are some minor comments I feel the authors can work on in this manuscript.

a) With respect to the the two tables , the authors should provide a short description as a table legend . In addition the table 1 is split with the last row separated.

b) additionally it would be best if the font sizes and style be uniform across both the table legends and the figure legends.

c) With respect to the figure 2, where the authors describe the role of KLF10 in cancers, the image looks cluttered and the text needs to be uniform across all the segments of the image. Furthermore, presenting this image in a different format with the suggestive working mechanism of KLF10 in pancreatic cancer as opposed to other cancers or similar to other cancers would be ideal to the readers. 

Author Response

The authors of the manuscript entitled "Krüppel-like Factor 10 as a Prognostic and Predictive Biomarker of Radiotherapy in Pancreatic Adenocarcinoma", Yi-Chih Tsai et al have done a commendable job in summarizing the role of SMAD4 and KLF10 in PDAC as well as their association with TGFB. The review however, directs towards the need for future trials to establish such biomarkers. The manuscript is well written and comprehensive. However, there are some minor comments I feel the authors can work on in this manuscript.

1. With respect to the the two tables , the authors should provide a short description as a table legend . In addition the table 1 is split with the last row separated.

Ans: Thank you for the comment. We added short description to each tables accordingly. We also revised the format of table 1.

2. additionally it would be best if the font sizes and style be uniform across both the table legends and the figure legends.

Ans: Thank you for the comments. We unified the font sizes and style of legends of tables and figures especially of figure 2 accordingly.

3. With respect to the figure 2, where the authors describe the role of KLF10 in cancers, the image looks cluttered and the text needs to be uniform across all the segments of the image. Furthermore, presenting this image in a different format with the suggestive working mechanism of KLF10 in pancreatic cancer as opposed to other cancers or similar to other cancers would be ideal to the readers. 

Ans: Thank you for the comment. We revised figure 2 in a more comprehensive, clear and mechanistic way according to your suggestion.